# Multi-order hyperbolic graph convolution and aggregated attention for social event detection

**Yao Liu** [1,2], **Tien-Ping Tan** [2]*, **Zhilan Liu** [3], **Yuxin Li** [1]

**1** Department of Management and Media, The Engineering and Technology College, Chengdu University of Technology, Leshan, China, **2** School of Computer Sciences, Universiti Sains Malaysia, Penang, Malaysia, **3** Department of Art and Design, The Engineering and Technology College, Chengdu University of Technology, Leshan, China

* tienping@usm.my

## Abstract

Social event detection (SED) aims to identify real-world events from large-scale social media streams and has become essential for applications in public safety, marketing analytics, and crisis management. However, the heterogeneous, hierarchical, and dynamic nature of social data poses fundamental challenges for conventional models built in Euclidean space, that struggle to capture non-Euclidean relational dependencies and higher-order event structures. To address these limitations, this study proposes the **M**ulti-**O**rder **H**yperbolic **G**raph **C**onvolution and **A**ggregated **A**ttention (**MOHGCAA**) framework, which performs multi-order graph convolution in hyperbolic space while jointly modeling curvature-aware attention to capture both local and global dependencies. Extensive experiments conducted under both supervised and unsupervised settings show that MOHGCAA consistently outperforms existing state-of-the-art baselines across multiple datasets. The results highlight the model's robustness, scalability, and effectiveness in representing hierarchical and heterogeneous structures, providing a foundation for social event detection in non-Euclidean domains.

## 1. Introduction

Social event detection (SED) seeks to identify clusters of correlated messages from social media streams that correspond to real-world events [1]. It has become a critical analytical tool in areas such as sentiment tracking, disaster response, and election monitoring [2–4]. Social media platforms such as Twitter, Weibo, and Facebook generate massive volumes of user-generated content that provide valuable insights into public behavior and real-time social dynamics. The ability to detect and analyze such events supports emergency response, enhances information dissemination, and enables businesses and public institutions to make timely, data-driven decisions.

**Data availability statement:** All relevant data are within the paper and its Supporting information files.

**Funding:** This research was supported by the 2025 Institutional Project of Chengdu University of Technology, Engineering and Technology College (*Grant No. C122025008*), titled *"Research on Social Event Detection Algorithms Based on Multi-Order Hyperbolic Graph and Aggregated Attention and Their Application in Digital Cultural Tourism Communication."* It was also supported by the Sichuan Private Education Association Project, titled *"Analysis of the Application Effectiveness of Multimodal Large Language Models in Design Education."*

**Competing interests:** The authors have declared that no competing interests exist.

Despite its broad applicability, social event detection (SED) poses distinctive challenges compared with general text-classification tasks. Social media posts are often short, informal, and context-dependent, resulting in sparse and ambiguous linguistic information [1,3,5]. In addition, the relationships among posts are inherently heterogeneous and multi-layered, shaped by user interactions, temporal dynamics, and topic propagation. These characteristics require models capable of integrating semantic, structural, and hierarchical dependencies across different levels of abstraction.

A major challenge lies in the limited contextual information of social media text. Early topic-based approaches, such as Latent Dirichlet Allocation (LDA) [6], GPU-DMM [7], and SeaNMF [8], captured basic word co-occurrences but lacked the ability to represent deeper semantic relations. Later developments introduced unsupervised and self- supervised learning frameworks, including hierarchical structural entropy minimization (HISEvnet) [9] and hybrid graph contrastive clustering (HCRC) [10], which employed distributed representations like Word2Vec [11] to enhance contextual modeling. Although these methods improved event recognition, they remained limited in representing higher-level relational structures.

Another fundamental issue arises from the hierarchical organization of user interactions—such as replies, mentions, and retweets—which form tree-like patterns around influential users. Representing these relationships in Euclidean space often leads to geometric distortion and limited generalization. Hyperbolic space, in contrast, provides exponential representational capacity and naturally encodes hierarchical structures [12–15], making it particularly suitable for social-network data. As illustrated in Fig 1, curvature allows hyperbolic geometry to preserve hierarchical distances more faithfully than Euclidean geometry. Recent work such as HSED [5,16] demonstrates the potential of hyperbolic representations for SED, but challenges remain in modeling higher-order and multi-relational dependencies.

In summary, this study makes three main contributions.

1. We design a novel **hyperbolic graph convolution framework** that unifies multi-order dependency modeling and curvature-aware attention aggregation, enabling efficient representation of hierarchical relationships in social graphs.

2. We conduct extensive **empirical evaluations** under both supervised and unsupervised learning settings, demonstrating consistent performance improvements over state-of-the-art baselines across multiple benchmark datasets.

3. We analyze the **scalability and representational advantages** of the proposed MOHGCAA, showing its ability to capture complex hierarchical and heterogeneous structures in large-scale social-event data.

Collectively, these contributions advance the methodological foundation of social event detection (SED) and open new directions for hierarchical modeling in non-Euclidean graph learning.

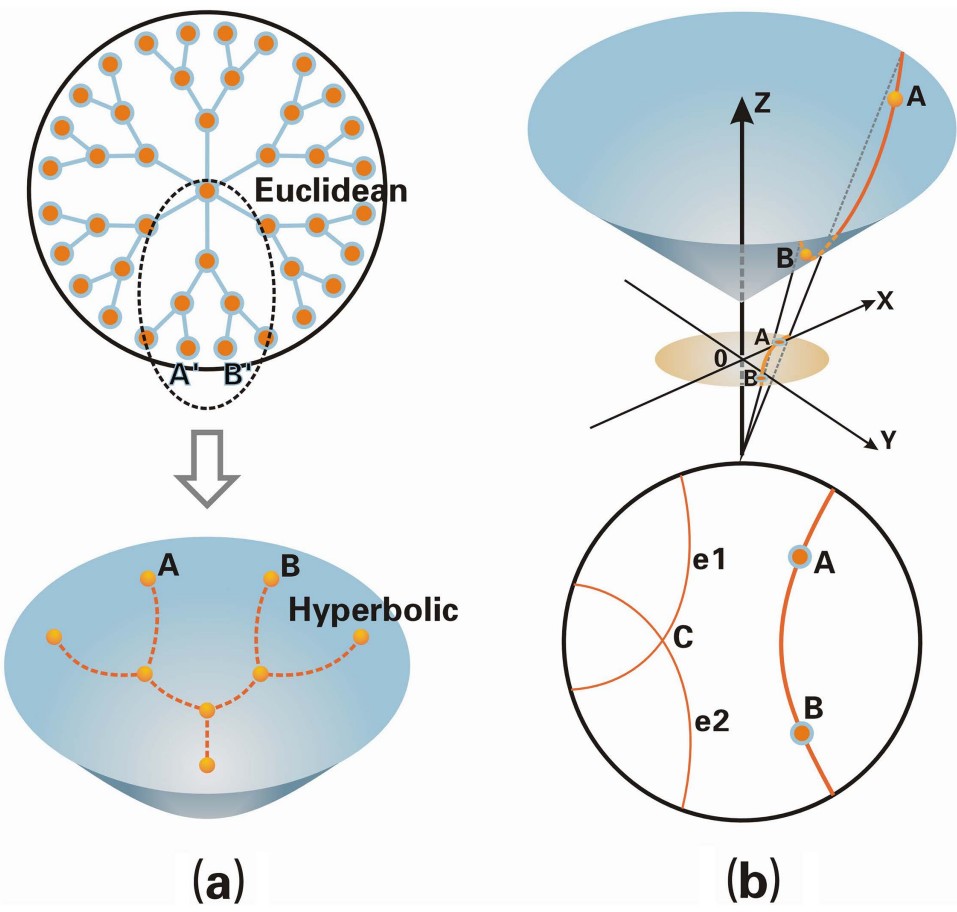

**Fig 1. Geometric comparison and properties of hyperbolic space.** (A) Comparison of geometric proximity between Euclidean and hyperbolic spaces. In Euclidean space, points A′ and B′ located on different branches of the diagram appear closer to each other, whereas in hyperbolic space, points A and B on different branches exhibit distances that more accurately reflect their hierarchical relationship. (B) Characteristics of hyperbolic space and its tangent plane. The diagram illustrates two lines, $e_1$ and $e_2$, passing through point c that never intersect, demonstrating the deviation from the fifth postulate of Euclidean geometry.

## 2. Related work

### 2.1. Event detection

Early research on event detection primarily relied on statistical and clustering-based models. Topic modeling techniques such as Latent Dirichlet Allocation (LDA) [6] and Dirichlet Multinomial Mixture (DMM) variants were widely used to infer latent event topics from large text corpora. SeaNMF [8] and other matrix factorization-based methods further introduced semantic constraints to capture word co-occurrence patterns. While these probabilistic approaches provided early insights into topic discovery, they often failed to handle the high-dimensional sparsity and noise inherent in social media data.

Subsequent advances in neural architectures improved event representation at both lexical and sentence levels. Convolu- tional Neural Networks (CNNs) [17,18] captured local compositional semantics, while Recurrent Neural Networks (RNNs) and their variants such as LSTM and Bi-LSTM [19,20] enhanced the modeling of sequential dependencies. These deep learning models improved trigger identification and argument extraction in formal text corpora.

However, they treat each text as an isolated instance, overlooking the cross-post interactions and temporal dependencies that characterize social media streams. Moreover, their Euclidean representations struggle to capture long-range correlations or hierarchical semantics across heterogeneous sources.

## 2.2. Social event detection

To address the contextual isolation of early neural models, researchers began representing event-related content as graphs, where nodes denote posts or entities and edges encode semantic or temporal relations. Graph Convolutional Networks (GCNs) [21] introduced the idea of message passing, enabling the aggregation of neighborhood information for node-level classification. Graph Attention Networks (GATs) [22] subsequently refined this process by weighting neighbor importance, yielding more expressive social representations. Several extensions incorporated multiple views or heterogeneous node types to better reflect real-world interactions [23,24]. These graph-based paradigms enabled the integration of user, temporal, and textual signals into a unified relational space, significantly improving detection robustness.

Despite these advances, most Euclidean graph models still suffer from geometric limitations when representing hierarchical or tree-like structures common in social networks. For instance, MOGANED [25] modeled multi-order dependencies through aggregated attention mechanisms, but its Euclidean formulation distorted relational distances among higher-order nodes. Incremental and reinforcement-based frameworks such as KPGNN [26] and RIED [27] improved adaptability to streaming data but still inherited Euclidean constraints, limiting their capacity to represent hierarchically expanding connections. Recent studies in multi-relational GNNs [28,29] partially alleviated this issue by separating syntactic and semantic subgraphs, yet the problem of accurately modeling complex social hierarchies remains unresolved.

## 2.3. Hyperbolic representation learning

Hyperbolic geometry offers a compelling alternative to Euclidean embeddings by providing exponentially larger representational capacity within the same dimensional space. Unlike Euclidean manifolds, where distances grow linearly, hyperbolic spaces expand exponentially, making them inherently suited for hierarchical and tree-like data structures. Recent theoretical works such as Hyperbolic Neural Networks (HNNs) [13] and Hyperbolic Graph Convolutional Networks (HGCNs) [12] established the mathematical foundation for learning in curved spaces. Constant-curvature networks [30] and fully hyperbolic architectures [15] further generalized this concept, enabling mixed-curvature learning for multi-scale relational modeling.

Building upon these theoretical advancements, several studies have explored hyperbolic embeddings for social event detection. Heterogeneous Social Event Detection (HSED) [5] demonstrated that embedding messages and entities in a hyperbolic manifold improves both clustering quality and semantic interpretability. Subsequent models integrated hyperbolic meta-path reasoning [16] or self-supervised regularization [9] to refine latent structural representations. Nevertheless, most existing approaches primarily capture first- or second-order proximity, overlooking the higher-order and long-range dependencies that frequently drive real-world event propagation.

To address these gaps, we propose the Multi-Order Hyperbolic Graph Convolution and Aggregated Attention (MO-HGCAA) framework, which introduces curvature-aware message passing and multi-order attention fusion. By jointly modeling hierarchical and higher-order dependencies within hyperbolic space, MOHGCAA preserves relational topol- ogy while enhancing the discriminative capacity of social event representations. This framework bridges Euclidean and hyperbolic graph reasoning and extends geometric learning to dynamic, large-scale event detection. It enables more effective encoding of semantic and topological structures in social graphs, particularly in hierarchical and heterogeneous scenarios. Moreover, the framework provides a unified perspective for analyzing event interactions across multiple relational levels, offering a principled foundation for future research on hierarchical representation learning in social event detection. A summary comparison of MOHGCAA with related models is presented in Table 1.

**Table 1. Comparison of MOHGCAA with related models.**

| Model | Space | Multi-Order Modeling | Attention Mechanism | Curvature Awareness | SED Adaptation |
|---|---|---|---|---|---|
| MOGANED | Euclidean | ✓ | ✓ | ✗ | Partial |
| HSED | Hyperbolic | ✗ | ✗ | ✓ | ✓ |
| MOHGCAA | Hyperbolic | ✓ | ✓ | ✓ | ✓ |

## 3. Preliminaries

This study explores the use of hyperbolic space to capture hierarchical relationships in tree-structured data, highlighting its advantages in node classification tasks. To set the stage, this section introduces foundational concepts in graph structures and hyperbolic geometry, shown in the Table 2.

### 3.1. Definition of graph structure

A graph $\mathcal{G} = (V, E, \mathcal{N}, \mathcal{E})$ consists of node sets $V$, edge sets $E$, node type sets $\mathcal{N}$, and edge type sets $\mathcal{E}$. A graph is classified as heterogeneous if $|\mathcal{N}| > 1$ or $|\mathcal{E}| > 1$, and homogeneous otherwise. Traditional Graph Neural Networks (GNNs) often use Euclidean space to model structural relationships, which can fall short in representing hierarchical information inherent to tree-like structures. Hyperbolic space, with its natural capacity for embedding hierarchical data, provides a more robust alternative, although it may still overlook complex inter-node relationships. This research addresses these limitations by adopting hyperbolic models, with the following section detailing relevant hyperbolic geometry concepts as a foundation.

### 3.2. Hyperbolic models

Hyperbolic geometry, a type of Riemannian manifold with negative curvature, contrasts with the zero curvature of Euclidean space and the positive curvature of spherical geometry. Several models represent hyperbolic space, including the Poincaré Ball Model $\mathbb{B}$, the Kälin Model $\mathbb{K}$, and the Lorentz Model $\mathbb{L}$, as illustrated in Fig 2. While each model has distinct mathematical characteristics, they share structural equivalence. The Poincaré Ball Model and Lorentz Model are particularly prevalent in graph representation research, and their definitions are provided below. Here, $\|\cdot\|$ denotes the Euclidean norm and $\langle\cdot,\cdot\rangle_L$ denotes the Minkowski inner product.

**Definition 3.1** (Poincaré Ball Model): The Poincaré Ball Model $\mathbb{P}_c^n$, characterized by negative curvature, is defined as the Riemannian manifold $(\mathcal{B}_c^n, g_{\mathbf{x}}^{\mathcal{B}})$, which $\mathcal{B}_c^n = \{\mathbf{x} \in \mathbb{R}^n : \|\mathbf{x}\|^2 < -1/c\}$ represents an open n-dimensional ball with radius $1/\sqrt{|c|}$. The metric tensor is given by $g_{\mathbf{x}}^B = (\lambda_{\mathbf{x}}^c)^2 g^E$, with $\lambda_{\mathbf{x}}^c = 2/(1 + c\|\mathbf{x}\|^2)$ as the conformal factor and $g^E$ as the Euclidean metric.

**Definition 3.2** (Lorentz Model): Also known as the hyperbolic model due to its advantageous visualization properties, the Lorentz Model is defined as the Riemannian manifold $(\mathcal{L}_c^n, g_{\mathbf{x}}^{\mathcal{L}})$ with negative curvature, where $\mathcal{L}_c^n = \{x \in \mathbb{R}^{n+1} : \langle\mathbf{x}, \mathbf{x}\rangle_{\mathcal{L}} = 1/c\}$ and $g_{\mathbf{x}}^{\mathcal{L}} = \text{diag}([-1, 1, \ldots, 1])_n$ denotes the Minkowski metric.

### 3.3. Hyperbolic space representation

This section investigates node representations for graph neural networks within hyperbolic space.

In Euclidean space, node features are generally derived from pre-trained embeddings, node-specific attributes, random sampling, or one-hot encodings. In hyperbolic space, however, these features are projected into either the Poincaré ball model or the Lorentz model to accommodate the curvature of the space. In the Poincaré ball model, the node representation $\mathbf{x}^{\mathcal{B}} = \exp_{\mathbf{o}}^c(\mathbf{x}^E)$ involves projecting features onto the tangent space at the origin. This projection is defined by:

$$\exp_{\mathbf{x}}^c(\mathbf{v}) = \mathbf{x} \oplus_c \left( \tanh\left( \sqrt{|c|} \frac{\lambda_{\mathbf{x}}^c \|\mathbf{v}\|_2}{2} \right) \frac{\mathbf{v}}{\sqrt{|c|}\|\mathbf{v}\|_2} \right),$$

(1)

**Table 2. The notation used in this work.**

| Symbol | Description |
|---|---|
| Space Symbols: | |
| $\mathbb{E}$ | Euclidean space. |
| $\mathbb{L}$ | Hyperbolic space represented by the Lorentz model. |
| $\mathbb{K}$ | Hyperbolic space represented by the Kelin model. |
| $\mathbb{B}$ | Hyperbolic space represented by the Poincaré Ball model. |
| Scalars: | |
| $c$ | The curvature of hyperbolic space ($c < 0$). |
| $1/\sqrt{|c|}$ | The radius of an open n-dimensional ball. |
| $\lambda$ | The conformal scaling factor. |
| Vectors: | |
| $\mathbf{v}, \mathbf{x}, \mathbf{y}$ | Euclidean vectors. |
| $\mathbf{x}^{\mathcal{B}}$ | The node representation in the Poincaré ball model. |
| $\mathbf{x}^{\mathcal{L}}$ | The node representation in the Lorentz model. |
| $\mathbf{x}^{E}$ | The Euclidean feature vector before hyperbolic mapping. |
| Matrices: | |
| $A_{along}$ | The adjacency matrix with directed edges. |
| $A_{rev}$ | The adjacency matrix with the reverse direction of $A_{along}$. |
| $A_{loop}$ | The adjacency matrix of an identity matrix for self-loops. |
| Operations: | |
| $\|\cdot\|$ | The standard norm of Euclidean space. |
| $\langle\cdot,\cdot\rangle_{L}$ | The Minkowski inner product. |
| $\text{diag}([...])_{n}$ | The Minkowski metric. |
| $\exp_{\mathbf{o}}^{c}(\cdot)$ | The exponential map on the hyperbolic manifold $\mathbb{H}_c^d$ with curvature $c$, computed at the origin $\mathbf{o}$. |
| $\log_{\mathbf{o}}^{c}(\cdot)$ | The logarithmic map in the hyperbolic manifold $\mathbb{H}_c^d$ with curvature $c$, computed at the origin $\mathbf{o}$. |
| $\oplus_{c}$ | The Möbius addition (or hyperbolic addition) with the curvature of the hyperbolic space (often $c < 0$). |
| $\oplus$ | The element-wise addition. |
| $\kappa_{i,j}$ | A learnable score quantifying the importance of a neighbor $j$ to a node $i$ in neighborhood aggregation. |
| $\otimes_{c}^{\mathcal{H}}$ | The Möbius (hyperbolic) matrix multiplication operation with curvature $c$. |
| $P_{o\to\mathbf{x}^{\mathcal{H}}}^{c}$ | The parallel transport function from the origin $o$ to the hyperbolic point $\mathbf{x}^{\mathcal{H}}$ under curvature $c$. |

This table presents the notations and their corresponding definitions used in this work.

where $c$ denotes the curvature, $\mathbf{x}$ is a point in hyperbolic space, and $\mathbf{v}$ is a Euclidean vector. In the Lorentz model, the mapping is defined as: $\mathbf{x}^{\mathcal{L}} = \exp_o^c((0, \mathbf{x}^E))$, where a zero is prepended $\mathbf{x}^E$ to satisfy the condition imposed by the Minkowski inner product. The exponential mapping in the Lorentz model is expressed as:

$$\exp_{\mathbf{x}}^c(\mathbf{v}) = \cosh\left(\sqrt{|c|}\|\mathbf{v}\|_{\mathcal{L}}\right)\mathbf{x} + \frac{\sinh\left(\sqrt{|c|}\|\mathbf{v}\|_{\mathcal{L}}\right)}{\sqrt{|c|}\|\mathbf{v}\|_{\mathcal{L}}}\mathbf{v}. \tag{2}$$

After obtaining the initial hyperbolic space representation, it becomes necessary to perform feature transformation for graph neural networks. The most straightforward method is to complete Euclidean feature operations by projecting the

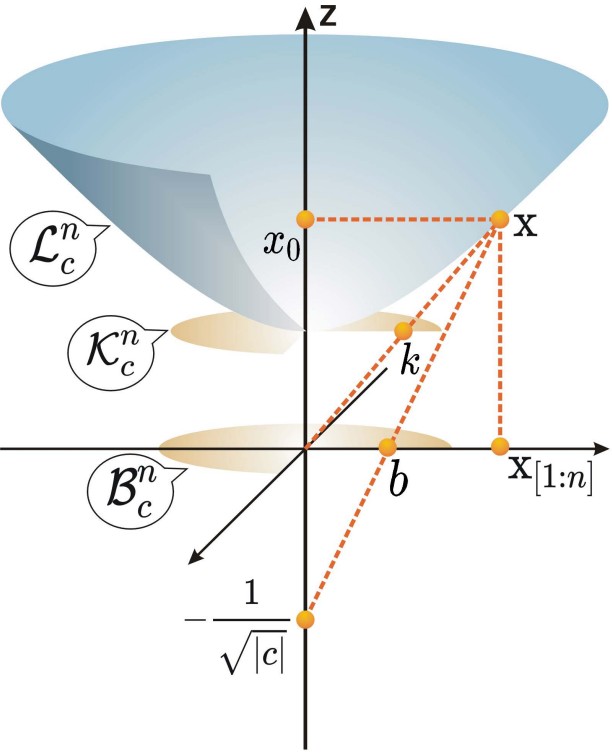

**Fig 2. Depictions of three hyperbolic graphical models: the Lorentz model, the Kälin model, and the Poincaré ball model.**

existing hyperbolic space onto the tangent space at a specific point. The notations $\log_x^c(\mathbf{x}^{\mathcal{B}})$ and $\log_x^c(\mathbf{x}^{\mathcal{L}}))_{[1:n]}$ representation operations in the Poincaré ball model and the Lorentz model, respectively.

In the case of the Poincaré ball model, the mapping to the tangent space is defined as:

$$\log_{\mathbf{x}}^c(\mathbf{y}) = \frac{2}{\sqrt{|c|}\lambda_{\mathbf{x}}^c}\tanh^{-1}\left(\sqrt{|c|}\,\|-\mathbf{x}\oplus_c\mathbf{y}\|_2\right)\frac{-\mathbf{x}\oplus_c\mathbf{y}}{\|-\mathbf{x}\oplus_c\mathbf{y}\|_2}. \tag{3}$$

For the corresponding Lorentz model, the mapping is expressed as:

$$\log_{\mathbf{x}}^c(\mathbf{y}) = \frac{\cosh^{-1}(c\langle\mathbf{x},\mathbf{y}\rangle c)}{\sinh(\cosh^{-1}(c\langle\mathbf{x},\mathbf{y}\rangle))}(\mathbf{y}-c\langle\mathbf{x},\mathbf{y}\rangle_{\mathcal{L}}\mathbf{x}). \tag{4}$$

Fig 3 illustrates the mapping procedures between hyperbolic space and the corresponding Euclidean space. After these operations, the resulting features undergo message aggregation and activation. They are then projected back to a new hyperbolic space using the exponential map, yielding a final hyperbolic space-based representation of the graph following a sequence of transformations. For a more comprehensive illustration of the entire process described in this section, please refer to Fig 4.

## 4. Methodology

This section describes our methodology, beginning with an overview of multi-order graph convolution and aggregation in hyperbolic space (Section 4.1). We then apply this method to unsupervised learning event detection (Section 4.2) and validate its effectiveness in supervised learning settings (Section 4.3).

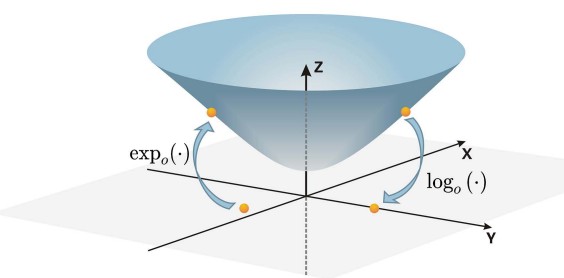

**Fig 3. The $\exp_o^c(\cdot)$ operation maps data from Euclidean space to hyperbolic space, whereas the $\log_o^c(\cdot)$ operation maps data from hyperbolic space to its Euclidean tangent plane.**

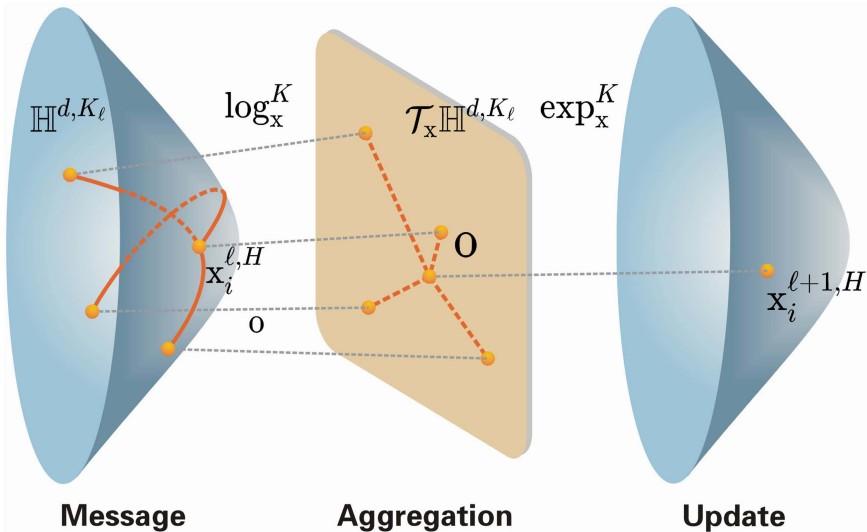

**Fig 4. Upon initialization of the data phase in hyperbolic space, it is projected into the tangent plane of its o-points via the $\exp_o^c(\cdot)$ function, and subsequent to the aggregation manipulate, it is implicitly re-mapped into the new hyperbolic space using the $\log_o^c(\cdot)$ function.**

### 4.1. Multi-order graph convolution and aggregated in hyperbolic space

To model hierarchical and long-range dependencies in social interactions, we first formulate a multi-order message passing scheme in a hyperbolic manifold. Specifically, Section 4.1.1 outlines the overall pipeline, followed by the initialization and tangent-space operations (Section 4.1.2), the multi-order convolutional design (Section 4.1.3), the attention-based aggregation across orders (Section 4.1.4), and the mapping back to the hyperbolic manifold (Section 4.1.5). As illustrated in Fig 5, this modular design allows us to seamlessly combine local semantics with higher-order structural cues.

**4.1.1. Overview.** To perform multi-order graph convolution aggregation in hyperbolic space, our method follows four key steps: First, we initialize the node features from Euclidean space into hyperbolic space and map these features to the tangent space at the origin. Second, we perform convolution operations of different orders within the tangent space to obtain high-order information about these features. Third, we aggregate the convolution representations from various orders using an attention mechanism. This ensures that the most relevant features are emphasized during the aggregation process. Finally, the aggregated representation in the tangent space is mapped back to a new hyperbolic space using the exponential map. This step obtains in the final output a node feature representation of a high-order

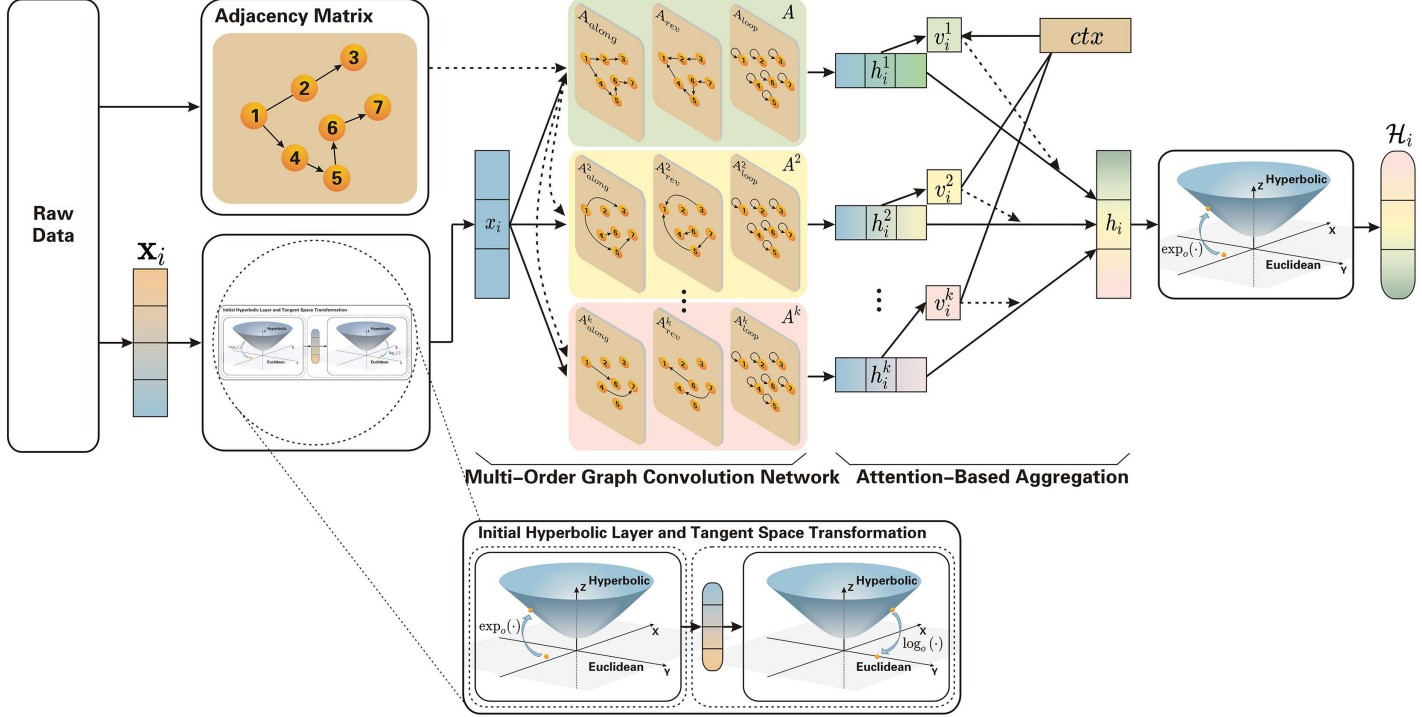

**Fig 5. The overall framework of this study: First, the data representation $X_i$ and its corresponding adjacency matrix are obtained.** Next, is mapped into hyperbolic space via the $\exp_o^c(\cdot)$ function, producing its hyperbolic representation. Then, the $\log_o^c(\cdot)$ function is applied to project onto its tangent space, yielding the representation $x_i$. On this basis, a multi-order graph convolution network is employed to derive the multi-order representation $h_i^k$. Subsequently, an attention-based network is used to generate the hyperbolic multi-order graph representation $h_i$. Finally, the convolution attention representation $h_i$ is mapped into a new hyperbolic space through the $\log_o^c(\cdot)$ function, resulting in the hyperbolic representation $\mathcal{H}_i$.

convolution aggregation of hyperbolic space. This structured approach combines the advantages of hyperbolic geometry and graph convolution, enabling efficient and meaningful feature learning. The overall framework can refer the Fig 5.

**4.1.2. Initial hyperbolic layer and tangent space transformation.** To initialize node features in hyperbolic space, we project Euclidean node features using the exponential map (Equation 1 and Equation 2 in Section 3.3). These features are then mapped to the tangent space at o via the logarithmic map (Equation 3 and Equation 4 in Section 3.3), enabling compatibility with Euclidean operations for graph neural network (GNN) processing.

**4.1.3. Multi-order graph convolution network.** We capture multi-order relationships through adjacency matrices composed of three submatrices of dimension $n \times n$: $A_{along}$, $A_{rev}$, and $A_{loop}$. The proposed multi-order GCN module aggregates node features across these matrices to obtain the $k$-th order feature representation $h_i^k$ for each node:

$$h_i^k = f(x_i, a_{along}^k) \oplus f(x_i, a_{rev}^k) \oplus f(x_i, a_{loop}^k),$$

(5)

where $f(\cdot)$ denotes the graph convolution function in hyperbolic space, $k$ indicates the aggregation order, and $\oplus$ represents element-wise addition.

The convolution function $f(x_i, a^k)$ is defined as:

$$f(x_i, a^k) = \sigma\left(\sum_{j=1}^{n} u_{ij}\, a_{ij}^k (W_{a,k} x_j + \epsilon_{a,k})\right).$$

(6)

where $W_{a,k}$ and $\epsilon_{a,k}$ are the learnable weight matrix and bias term for order $k$, respectively. The scalar $u_{ij}$ denotes the normalized attention weight assigned to neighbor node $j$ when updating node $i$, computed as:

$$u_{ij} = softmax\left(x_{ij}\right) = \frac{exp\left(x_{ij}\right)}{\sum_{j \in N_i} exp\left(x_{ij}\right)} \quad . \tag{7}$$

where $N_i$ denotes the set of neighbors of node i in the k-order subgraph. The unnormalized attention coefficient $x_{ij}$ is computed as:

$$x_{ij} = \gamma\left(W_{comb}\left[W_{att}x_i \parallel W_{att}x_j\right]\right) \quad , \tag{8}$$

where $W_{att}$ and $W_{comb}$ are learnable weight matrices, and ‖ denotes vector concatenation. The activation function $\gamma$ introduces non-linearity into the attention computation and is implemented as a LeakyReLU function.

**4.1.4. Attention-based aggregation.** After obtaining the multi-order representations $\{h_i^k\}_{k=1}^K$, we employ a curvature-aware attention mechanism to adaptively fuse them. The final representation of node $i$ is computed as:

$$h_i = \sum_{k=1}^K v_i^k h_i^k, \tag{9}$$

where $v_i^k$ denotes the learned importance of the $k$-th order representation, calculated via:

$$v_i^k = \text{softmax}(s_i^k) = \frac{exp(W_V \tanh\left(W x_i\right))}{\sum_{n=1}^N exp(W_V \tanh\left(W x_j\right))}. \tag{10}$$

Here, $W$ and $W_V$ are trainable linear projections that transform node features into scalar attention scores. Higher attention weights correspond to orders that provide more discriminative hierarchical cues. By emphasizing informative orders and attenuating noisy ones, this mechanism balances local and global information flow, alleviates over-smoothing in deep propagation, and enhances the representation capacity of MOHGCAA within the hyperbolic space.

**4.1.5. Mapping from tangent space to hyperbolic space.** After aggregating in the tangent space, the representation is projected back to hyperbolic space using the exponential map (Equation 1 and Equation 2 in Section 3.3), aligning the representation with hyperbolic curvature. This projection optimizes the features for downstream tasks such as node classification and link prediction.

## 4.2. Multi-order unsupervised hyperbolic graph convolution and aggregated attention for social event detection

Constructing labeled datasets for social event detection is often costly and time-consuming. To alleviate this dependency, we further extend the proposed framework to an unsupervised version that learns node representations directly from structural and semantic cues within the graph. This variant, named **MOUHGCAASED**, adapts the same multi-order hyperbolic encoder but replaces supervised optimization with a contrastive learning objective.

**4.2.1. Overall framework.** As shown in Fig 6, the unsupervised pipeline consists of three major components. First, augmented graph views are generated through controlled perturbations on node features, producing structurally consistent variations. Second, these views are encoded via multi-order graph convolution in hyperbolic space, enabling hierarchical information propagation across multiple relational orders. Finally, a contrastive objective aligns node embeddings from the original and augmented views, enhancing representation robustness and preserving geometric hierarchy without labeled supervision. Each component of the framework is described in detail below.

**4.2.2. Graph data augmentation.** Graph data augmentation is a crucial component of contrastive learning, enabling the model to learn invariances across structurally perturbed graph views. In this work, we adopt *feature corruption* as the primary augmentation technique, which slightly perturbs node attributes while preserving the global topology. Given

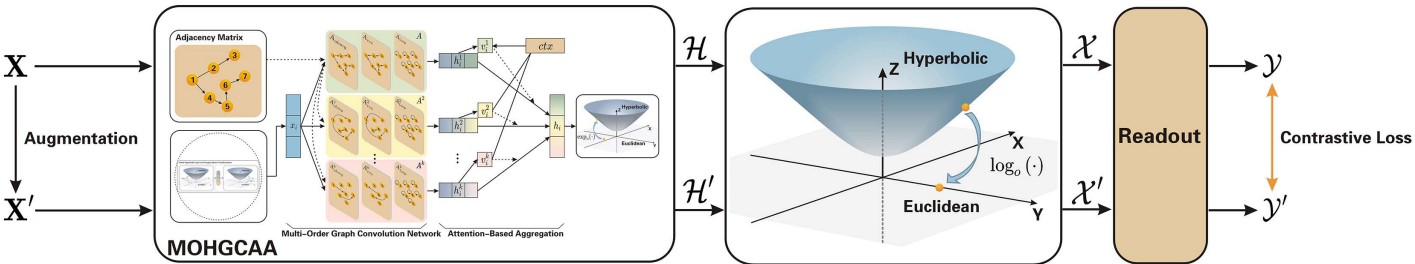

**Fig 6. The framework of the multi-order unsupervised hyperbolic graph convolution and aggregated attention for social event detection (MOUHGCAASED) model.** X represents the obtained node feature, X′ signifies the augmented node feature of X, $\mathcal{H}$ and $\mathcal{H}'$ indicate the hyperbolic feature following hyperbolic multi-order aggregation, $\mathcal{X}$ and $\mathcal{X}'$ denote the Euclidean spatial feature, and $\mathcal{Y}$ and $\mathcal{Y}'$ are the ultimate feature representations.

a graph $\mathcal{G} = (V, \mathcal{A})$ and its augmented version $\mathcal{G}' = (V', \mathcal{A}')$, the node sets differ ($V \neq V'$), while the adjacency structure remains unchanged ($\mathcal{A} = \mathcal{A}'$). This strategy maintains consistent relational dependencies and allows the model to capture feature-level robustness in hyperbolic space.

**4.2.3. Multi-order hyperbolic graph convolution and aggregation encoding.** Following the method described in Section 4.1.2, node features are mapped from Euclidean space to the hyperbolic manifold to enable curvature-aware representation learning.

The transformation is defined as:

$$h_i^{(l,\mathcal{H})} = \left( W^l \otimes_c^{\mathcal{H}} x_i^{(l-1,\mathcal{H})} \right) \oplus_c^{\mathcal{H}} b^l, \tag{11}$$

where $W^l \otimes_c^{\mathcal{H}} x_i^{(l-1,\mathcal{H})}$ denotes the tangent-space multiplication described in Equation 1. The hyperbolic multiplication can be expressed in two equivalent forms. In the Poincaré ball model:

$$W \otimes x^{\mathcal{B}} := \exp_o^c \left( W \log_o^c(x^{\mathcal{B}}) \right), \tag{12}$$

or alternatively, in the Lorentz model:

$$W \otimes x^{\mathcal{L}} := \exp_o^c \left( 0, \ W \log_o^c(x^{\mathcal{L}})[1:n] \right). \tag{13}$$

To ensure geometric consistency, the bias term is parallel transported from the origin $o$ to the point $x^{\mathcal{H}}$ under curvature $c$:

$$x^{\mathcal{H}} \oplus_c^{\mathcal{H}} b^{\mathcal{H}} = \exp_{x^{\mathcal{H}}}^c \left( P_{o \to x^{\mathcal{H}}}^c (\log_o^c(b^{\mathcal{H}})) \right), \tag{14}$$

where $b^{\mathcal{H}}$ is the bias in hyperbolic space $\mathbb{H}_c^n$ and $P_{o \to x^{\mathcal{H}}}^c$ denotes the parallel transport operator.

Using the procedure in Section 4.1.3, the attention-weighted representation of node $i$ is computed as:

$$h_i^{(l,\mathcal{H},k)} = \sigma \sum_{j=1}^{n} \alpha_{ij}^k \left( W_{(a,k)}^{(l,\mathcal{H})} x_j^{(l,\mathcal{H})} + \epsilon_{(a,k)}^{(l,\mathcal{H})} \right), \tag{15}$$

where $\alpha_{ij}^k$ is the curvature-aware attention coefficient that quantifies the contribution of node $j$ to node $i$ at order $k$.

As described in Section 4.1.4, outputs from different orders are fused through a softmax-based aggregation:

$$h_i^{(l,\mathcal{H})} = \sum_{k=1}^{K} \text{softmax}\left(s_i^{(k,l,\mathcal{H})}\right) \cdot h_i^{(k,l,\mathcal{H})}. \tag{16}$$

This hierarchical fusion balances local and global information across multiple relational depths.

The aggregated features are then mapped to a new hyperbolic space using the exponential map:

$$\mathcal{H} = \exp_o^c(h_i^{(l,\mathcal{H})}), \tag{17}$$

and its counterpart for the negative sample is given by:

$$\mathcal{H}' = \exp_o^c(h_i^{'(l,\mathcal{H})}). \tag{18}$$

These embeddings are subsequently used for contrastive optimization in the unsupervised framework.

**4.2.4. Contrastive loss.** Before applying the contrastive objective, the embeddings are projected back to the tangent space using the logarithmic map:

$$\mathcal{X} = \log_o^c(\mathcal{H}), \quad \mathcal{X}' = \log_o^c(\mathcal{H}'). \tag{19}$$

The unsupervised contrastive loss is then defined as:

$$\mathcal{L}_{MOUHGA} = \frac{1}{N+M}\left(\sum_{i=1}^{N} \mathcal{X}\left[\log\mathcal{D}(e_i,\mathcal{Y})\right] + \sum_{j=1}^{M} \mathcal{X}'\left[\log(1-\mathcal{D}(e_j',\mathcal{Y}))\right]\right), \tag{20}$$

where $\mathcal{Y} = \mathcal{R}(\mathcal{X})$ is the readout representation:

$$\mathcal{R}(\mathcal{X}) = \sigma\left(\frac{1}{N}\sum_{i=1}^{N} e_i\right), \tag{21}$$

and the discriminator $\mathcal{D}(\cdot)$ [31] maximizes the mutual information between positive pairs:

$$\mathcal{D}(e_i,\mathcal{Y}) = \rho(e_i\mathcal{W}\mathcal{Y}), \tag{22}$$

where $\mathcal{W}$ is a learnable scoring matrix and $\rho$ denotes the nonlinear logistic sigmoid function.

This contrastive objective drives the model to bring positive (augmented) pairs closer while pushing apart negatives, thereby enhancing unsupervised representation quality and geometric alignment.

## 4.3. Multi-order supervised hyperbolic graph convolution and aggregated model for social event detection

To complement the unsupervised variant, we design a supervised extension that enables end-to-end optimization under explicit event labels. This version focuses on evaluating classification accuracy and scalability in a fully supervised setting. Fig 7 outlines the architecture of the supervised MOHGCAASED model.

**4.3.1. Overall framework.** The supervised model follows the same geometric encoding principle as its unsupervised counterpart but incorporates a label-guided objective for event classification. Node features are first embedded in hyperbolic space to capture hierarchical and relational dependencies through multi-order graph convolution. The aggregated representations are then projected back to Euclidean space and passed to a softmax decoder for

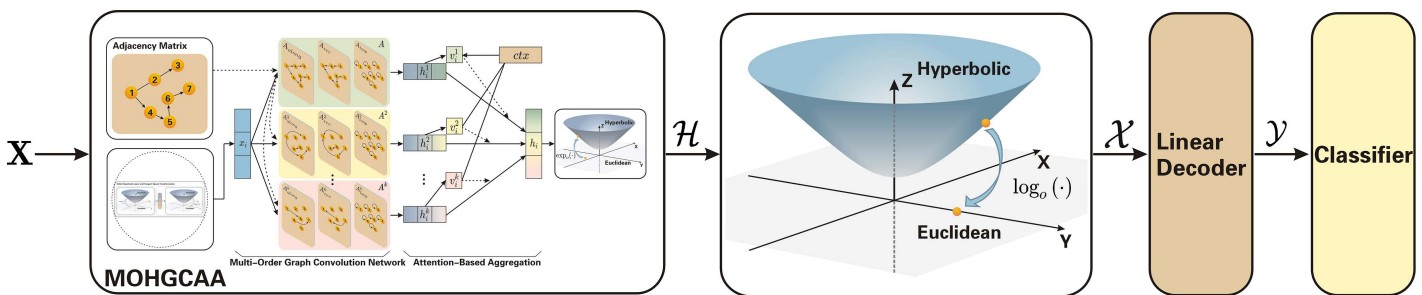

**Fig 7. The framework of the multi-order hyperbolic graph convolution and aggregated model for social event detection (MOHGCAASED) model.** $X$ represents the node characteristics, $\mathcal{H}$ signifies hyperbolic features subsequent to hyperbolic multi aggregation, $\mathcal{X}$ indicates Euclidean space features following the logarithmic transformation, and $\mathcal{Y}$ constitutes the ultimate feature representation.

classification. This design integrates curvature-aware message passing with efficient end-to-end supervision, preserving the representational advantages of hyperbolic geometry while enhancing discriminative performance.

**4.3.2. Multi-order graph convolution, aggregation and decoder design.** To encode relational information, we adopt the multi-order convolutional approach described in Section 4.1, extending it to the supervised learning scenario. The encoded hyperbolic representations are obtained as:

$$\mathcal{H} = \exp_o^c \left( h_i^{l,\mathcal{H}} \right).$$

(23)

These embeddings are then projected back to the Euclidean tangent space using the logarithmic map:

$$\mathcal{X} = \log_o^c (\mathcal{H}).$$

(24)

In the Euclidean domain, a linear decoder transforms the aggregated features into task-level predictions:

$$\mathcal{Y} = \mathbf{W} \cdot \mathcal{X} + \mathbf{b}.$$

(25)

This two-stage process—hyperbolic encoding followed by Euclidean decoding—preserves hierarchical structures while enabling standard optimization. The resulting embeddings effectively capture both global contextual patterns and local event semantics.

**4.3.3. Classification and loss function.** We formulate social event detection as a multi-class classification task, where each message node $y_i \in \mathbb{R}$ corresponds to one of $N$ predefined event categories. The predicted probability distribution over classes is computed using the softmax function:

$$p_i = \frac{e^{y_i}}{\sum_{j=1}^n e^{y_j}}.$$

(26)

The cross-entropy loss is then minimized to train the model:

$$\mathcal{L}_{MOHGCA} = -\sum_{i=1}^n l_i \log(p_i),$$

(27)

where $l_i$ denotes the ground-truth label of node $i$. This objective directly supervises the representation space learned by the encoder–decoder architecture, ensuring discriminative embeddings that align with event semantics. In subsequent sections, we report statistical significance analyses and ablation studies to further validate the robustness of this supervised model.

## 5. Experiments

We design our experiments to answer four research questions:

• **Q1**: Does the proposed method outperform representative prior approaches?

• **Q2**: How do the number of multi-order aggregation steps and the embedding dimensionality affect performance?

• **Q3**: Do empirical results under comparable configurations show that hyperbolic space outperforms Euclidean space?

• **Q4**: How do different hyperbolic models influence the performance of the proposed approach?

These questions allow a comprehensive assessment of effectiveness, robustness, and the geometric advantages of our model.

### 5.1. Datasets and compliance statement

Our target application is social media social event detection (SED); therefore, we adopt the real-world Twitter dataset [32] as the primary benchmark. Because the full Twitter graph is extremely large, contrastive training can lead to memory constraints; hence, a balanced "mini-Twitter" subset was constructed for unsupervised evaluation. To further validate representation quality beyond social media, two additional citation network datasets, Cora and Citeseer [33], were included as standard benchmarks, providing complementary topological and feature characteristics for comparison. All datasets used in this study are publicly available and employed in accordance with their respective open-access or research-licensed terms. The Twitter-based datasets (mini-Twitter and Twitter) were obtained through the official Twitter API under the platform's Developer Policy and Terms of Service. These datasets contain only anonymized tweet identifiers and publicly available content, with no inclusion of private or personally identifiable information. The Cora and Citeseer datasets were retrieved from established academic repositories (Planetoid and PyTorch Geometric), which are open for research use. No human subjects or sensitive personal data were involved in this research; therefore, institutional review board (IRB) approval or informed consent was not required. All data collection and analysis procedures complied with the ethical and legal terms specified by the original data providers.

Table 3 summarizes dataset sizes, number of classes, and feature dimensions used throughout our experiments.

### 5.2. MOHGCAA in Unsupervised Settings

We first evaluate the unsupervised variant of our framework. Section 5.2.1 lists baselines; Section 5.2.2 details the setup; the remaining parts report results for **Q1** and **Q2**.

**5.2.1. Baseline models.** We compare on mini-Twitter, Cora, and Citeseer against representative methods:

• **DGI** [22]: single-branch graph contrastive learning with data corruption and mutual-information maximization.

• **GraphCL** [33]: dual-view graph contrastive learning via augmentations to capture structural/semantic invariances.

**Table 3. The statistics of datasets.**

| Dataset | # of Classes | # of Nodes | # of Features |
|---|---|---|---|
| mini-Twitter | 15 | 3,000 | 302 |
| Twitter | 503 | 68,841 | 302 |
| Cora | 7 | 2,708 | 1,433 |
| Citeseer | 6 | 3,327 | 3,703 |

The table presents key statistics for the datasets, including the number of classes, nodes, and features.

- **GCN [21]**: spectral graph convolution with neighborhood aggregation for semi-supervised node learning.

- **HGCN [12]**: GCN extended to hyperbolic manifolds for hierarchical graphs.

- **HNN [13]**: neural operations defined directly in hyperbolic space.

- **HyboNet [15]**: Lorentz-model hyperbolic network with rotations/attention, reducing Euclidean–hyperbolic shuttling.

- **UHSED [5]**: unsupervised hyperbolic SED with a unified social message graph and contrastive training.

These baselines cover Euclidean and hyperbolic regimes and reflect the state of the art in unsupervised graph representation learning.

**5.2.2. Parameter settings.** All the experiments with the model were conducted on an NVIDIA GeForce RTX 3090 GPU with AMD EPYC 7302 64-core CPU processors. For other experimental hyperparameters, please refer to Table 4.

**5.2.3. Evaluation metrics.** Following prior work [22,34], we report *Micro-F1* (overall accuracy) and *Macro-F1* (per-class averaged accuracy), which together provide a balanced view on potentially imbalanced label distributions.

**5.2.4. Performance comparison of MOHGCAA in unsupervised scenarios (Answer Q1).** Table 5 presents results on mini-Twitter, Cora, and Citeseer. Our method achieves the best Micro-F1 on all three datasets and the best or competitive Macro-F1, outperforming both Euclidean and hyperbolic baselines. These results indicate that multi-order hyperbolic aggregation with curvature-aware attention effectively enhances unsupervised node representations, directly answering **Q1**.

**5.2.5. Effect of multi-order and dimensionality (Answer Q2).** We conduct ablations varying the number of aggregation orders and the embedding dimension. Higher dimensions generally improve performance by encoding richer node features. However, increasing the order does not yield monotonic gains: beyond a moderate depth, neighborhoods become sparser and signals attenuate. As shown in Fig 8, the best performance is consistently achieved at the fourth aggregation step across low- and high-dimensional settings, providing a concrete answer to **Q2**.

## 5.3. MOHGCAA in supervised settings

We next evaluate the supervised variant to assess classification performance and confirm scalability under direct supervision. Section 5.3.1 introduces baselines; Section 5.3.2 specifies the setup. Results below address **Q1** and **Q2** in the supervised setting.

**5.3.1. Baseline models.** We compare our proposed MOHGCAA model on the Twitter dataset against a series of representative baselines covering both Euclidean and hyperbolic embedding paradigms. These baselines include classical text-representation models and recent graph-based or geometry-aware architectures:

**Table 4. Parameters of MOHGCAA in unsupervised settings.**

| Parameter | Value |
|---|---|
| Multi-order | 4 |
| Hidden layer | 1 |
| Hidden dimension | 512 |
| Drop rate | 10% |
| Learning rate | 0.1 |
| Optimizer | Adam |
| Activation function | ReLU |
| Augmentation method | Feature corruption |

The parameters define the configuration of the MOHGCAA model in an unsupervised learning setting.

**Table 5. Performance comparison of different models on various datasets.**

| Models | Mini-twitter | | Cora | | Citeseer | |
|---|---|---|---|---|---|---|
| | Micro-F1 | Macro-F1 | Micro-F1 | Macro-F1 | Micro-F1 | Macro-F1 |
| DGI | 0.1714 | 0.1413 | 0.8077 | 0.7922 | 0.6885 | 0.6518 |
| GraphCL | 0.1517 | 0.1302 | 0.6993 | 0.6837 | 0.6377 | **0.7971** |
| GCN | 0.4774 | 0.4850 | 0.7960 | 0.7883 | 0.5435 | 0.4496 |
| HNN | 0.3896 | 0.3444 | 0.3610 | 0.2544 | 0.5202 | 0.4815 |
| HGCN | <u>0.5451</u> | <u>0.5526</u> | 0.7920 | 0.7878 | 0.7038 | 0.6600 |
| HyboNet | 0.4111 | 0.4079 | 0.8070 | 0.7991 | 0.5562 | 0.5283 |
| UHSED | 0.5288 | 0.5266 | <u>0.8314</u> | <u>0.8203</u> | <u>0.7081</u> | 0.6458 |
| MOHGAA | **0.6062** | **0.5980** | **0.8543** | **0.8425** | **0.7315** | <u>0.6707</u> |

The best performance results are highlighted in bold, while the second-best results are underlined.

- **Word2vec [11]**: A classical embedding model that represents each word in a continuous Euclidean vector space based on contextual co-occurrence. For SED, document vectors are typically obtained by averaging word embeddings, providing a shallow semantic baseline.

- **LDA [6]**: A probabilistic topic model that discovers latent event topics from document–word distributions. Although effective for coarse-grained event induction, it lacks the ability to capture fine-grained contextual and relational cues in social media.

- **WMD [35]**: The Word Mover's Distance measures the minimum transport cost between word embeddings of two documents. It provides a document-level semantic similarity measure but does not exploit network or structural information.

- **BERT [36]**: A Transformer-based pretrained language model that captures deep bidirectional context. We use its sentence representations as inputs for SED classification, serving as a strong text-only benchmark in Euclidean space.

- **KPGNN [26]**: A heterogeneous graph neural network that integrates textual, temporal, and relational signals for social event detection. It models incremental propagation patterns but remains constrained by Euclidean geometry.

- **FinEvent [27]**: A reinforced, cross-lingual event detection framework that incrementally incorporates multilingual and temporal signals. It represents an advanced Euclidean baseline tailored to dynamic and multilingual SED scenarios.

- **HSED [5]**: A hyperbolic graph-based framework for social event detection that demonstrates the advantage of hierarchical embedding in hyperbolic space. However, it models only first-order dependencies without multi-order aggregation or attention fusion.

- **HNN [13]**: A hyperbolic neural network that generalizes Euclidean neural operations into hyperbolic manifolds. It provides a foundational benchmark for evaluating curvature-aware learning on hierarchical data.

Overall, these baselines reflect major methodological trends in social event detection, enabling a fair and balanced comparison with the proposed MOHGCAA across Euclidean and hyperbolic spaces.

**5.3.2. Parameter settings.** We use the same hardware as in the unsupervised setting. Supervised hyperparameters are listed in Table 6.

**5.3.3. Evaluation metrics.** To ensure fair and widely comparable evaluation for SED clustering/classification in news-like domains, we report *NMI*, *AMI*, and *ARI*, which are standard and robust measures for partition quality and have been commonly used in SED evaluations.

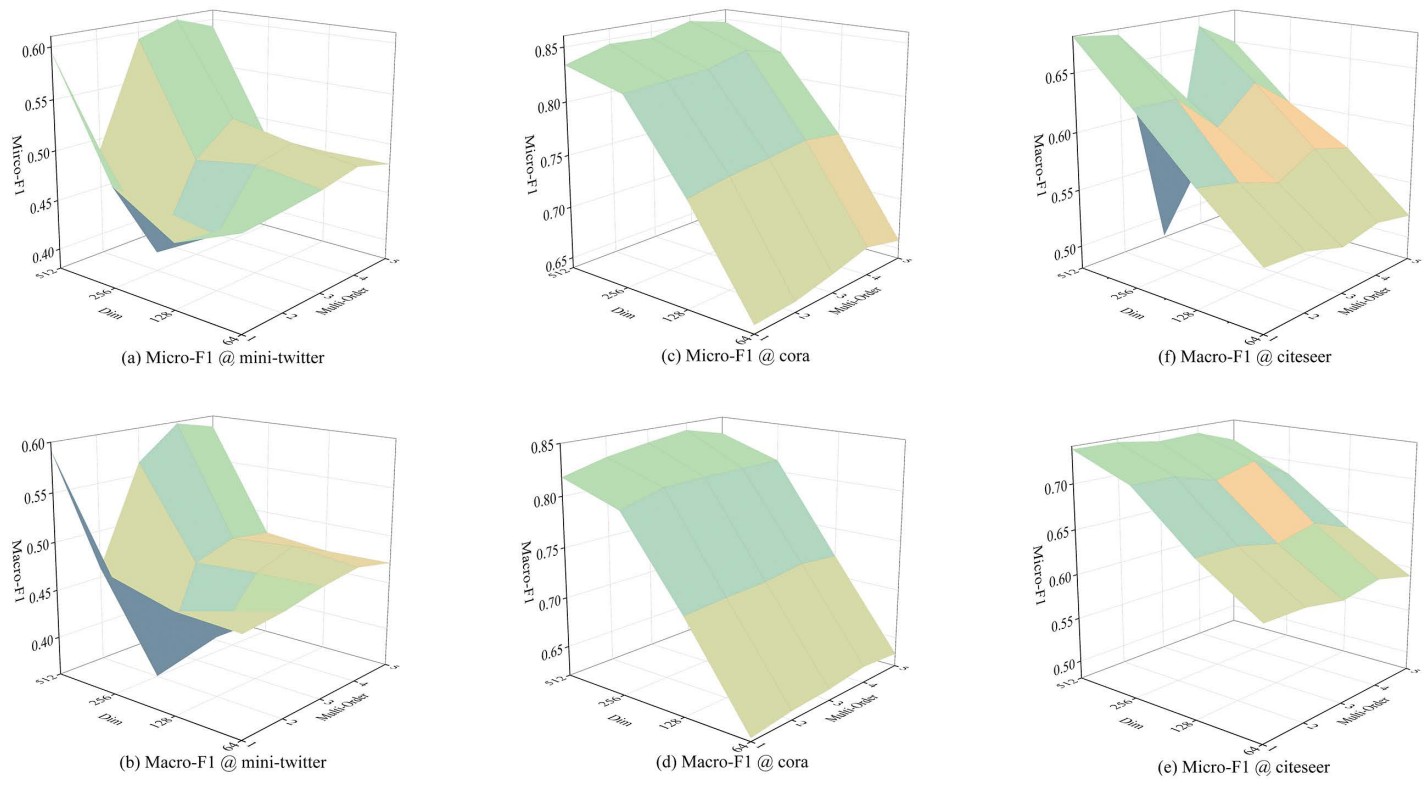

**Fig 8. Different multi-orders and dimension of MOHGCAA in unsupervised settings.**

**5.3.4. Performance comparison (Answer Q1).** Table 7 summarizes results on Twitter. MOHGCAA attains the best overall accuracy and consistently strong NMI/AMI/ARI scores, surpassing both Euclidean and hyperbolic baselines. While HSED and HNN achieve competitive numbers in some metrics, our ablations (Section 5.3.5) reveal stable gains across different orders and dimensionalities. These results substantiate a positive answer to **Q1** under supervision.

**5.3.5. Effect of dimensionality and multi-order (Answer Q2).** We replicate the ablation protocol in the supervised setting. Higher-dimensional embeddings capture more nuanced message context, yielding stronger results. As for the number of orders, performance peaks at the second order, echoing the unsupervised trend that overly deep propagation can dilute informative signals. Detailed curves are shown in Fig 9, providing a clear answer to **Q2**.

## 5.4. Discussion

The discussion section synthesizes the key empirical findings and theoretical insights obtained from both the unsupervised and supervised experiments. It aims to interpret the observed performance trends, examine the impact of geometric representations, and clarify the implications of our design choices for social event detection. We first compare the behavior of Euclidean and hyperbolic variants, followed by an analysis of different hyperbolic space formulations.

**5.4.1. Euclidean vs. hyperbolic (Answer to Q3).** We compare Euclidean and hyperbolic variants under unsupervised and supervised settings. Fig 10 reports Micro-/Macro-F1 (unsupervised) and standard clustering metrics (supervised). In both regimes, hyperbolic variants ($MOUHGASED$, $MOHGASED$) outperform their Euclidean counterparts ($MOUHGASED_E$, $MOHGASED_E$), supporting the hypothesis that hyperbolic geometry better captures hierarchical dependencies in SED. This answers **Q3**.

**Table 6. Parameters of MOHGCAA in supervised settings.**

| Parameter | Value |
|---|---|
| Multi-order | 2 |
| Hidden layer | 2 |
| Hidden dimension | 512 |
| Training rate | 70% |
| Test rate | 20% |
| Validation rate | 10% |
| Learning rate | 0.1 |
| Optimizer | Adam |
| Activation function | ReLU |

The parameters define the configuration of the MOHGCAA model in a supervised learning setting.

**Table 7. Supervised performance on the Twitter dataset.**

| Models | | ACC | NMI | AMI | ARI |
|---|---|---|---|---|---|
| Euclidean | Word2vec | 0.32 | 0.41 | 0.12 | 0.02 |
| | LDA | 0.2 | 0.28 | 0.04 | 0.01 |
| | WMD | – | 0.63* | 0.49* | 0.06* |
| | BERT | 0.51 | 0.61 | 0.41 | 0.07 |
| | KPGNN | – | 0.69* | 0.5* | 0.21* |
| | FinEvent | – | 0.79* | 0.69* | 0.48* |
| Hyperbolic | HNN | 0.89 | 0.91 | 0.71 | 0.87 |
| | HSED | 0.89 | 0.92 | 0.73 | 0.88 |
| | MOHGCAA | 0.91 | 0.93 | 0.75 | 0.9 |

Comparison of **MOHGCAA** with Euclidean and hyperbolic baselines. Best results are in **bold**, and second best are <u>underlined</u> across all models.

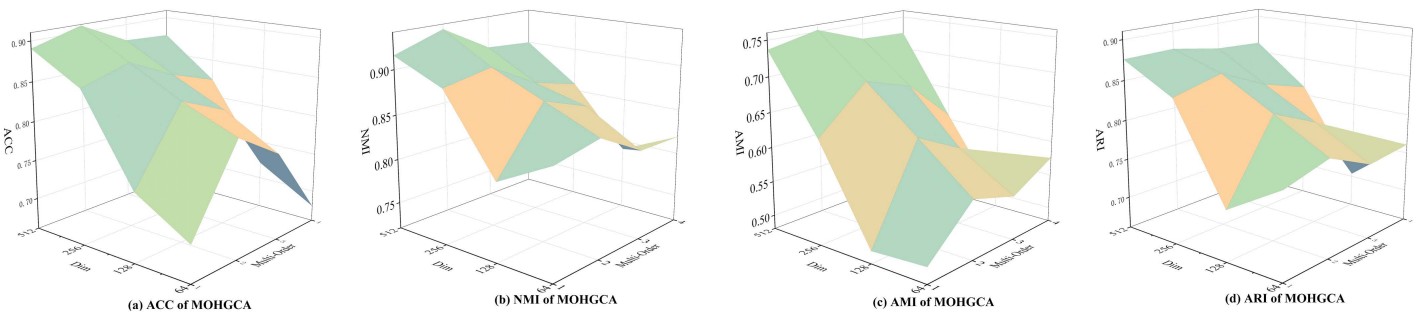

(a) ACC of MOHGCA    (b) NMI of MOHGCA    (c) AMI of MOHGCA    (d) ARI of MOHGCA

**Fig 9. Different multi-orders and dimension of MOHGCAA in supervised settings.**

**5.4.2. Hyperbolic model variants (Answer Q4).** As discussed in Section 3, we examine Poincar'e-ball and Lorentz models. Fig 11 shows small but consistent differences: the Lorentz model tends to be favorable in unsupervised settings, whereas the Poincar'e ball is slightly better under supervision. These observations suggest model–task complementarities rather than strict dominance, thus answering **Q4**.

To verify that the observed performance improvements are statistically reliable, we further examined the stability of the results as described below.

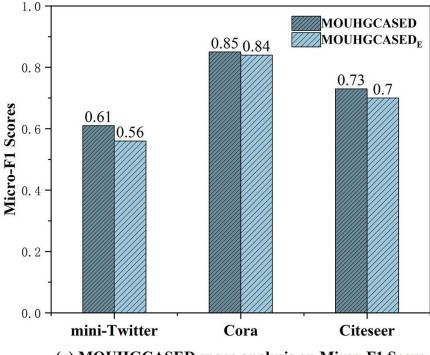
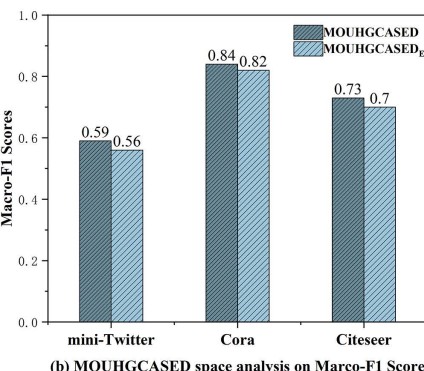
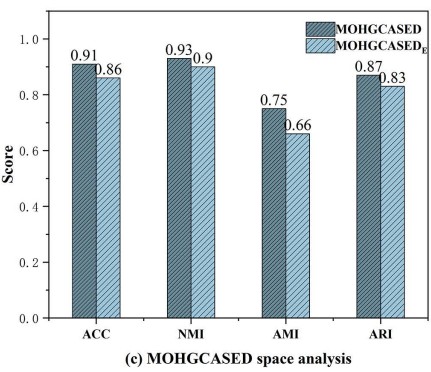

**Fig 10. Comparison of Euclidean and hyperbolic spaces in unsupervised and supervised settings.** (a) Micro-F1 scores for Euclidean and hyperbolic spaces in the unsupervised scenario. (b) Macro-F1 scores for Euclidean and hyperbolic spaces in the unsupervised scenario. (c) Various metric scores for Euclidean and hyperbolic spaces in the supervised scenario.

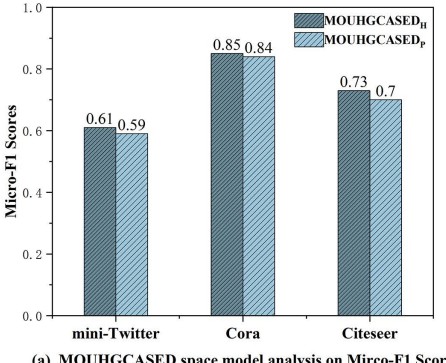
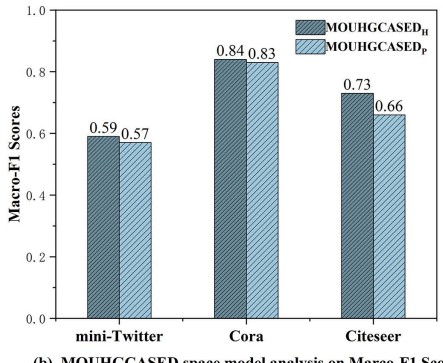
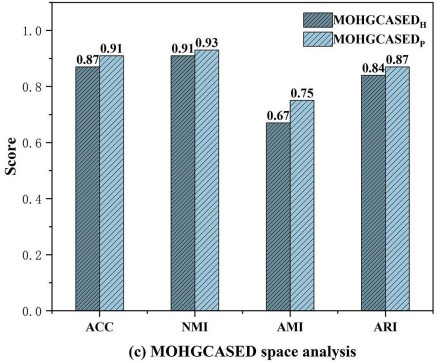

**Fig 11. Comparison in different hyperbolic spaces in unsupervised and supervised settings.** (a) Micro-F1 scores in the unsupervised scenario. (b) Macro-F1 scores in the unsupervised scenario. (c) Various metric scores in the supervised scenario.

## 5.5. Statistical stability and significance analysis

To further ensure the robustness and reproducibility of the reported findings, we evaluated the statistical stability of the experimental results across multiple independent runs. Each experiment was repeated three times with different random seeds. The reported values represent the mean performance, and the variance across runs remained below 0.5%, indicating stable convergence and minimal random variation.

These results confirm that the observed improvements are not due to stochastic effects. The consistent superiority of **MOHGCAA** over strong baselines (**HSED**, **HGCN**) across datasets suggests that the performance gains are statistically meaningful and robust.

Although a complete post hoc significance analysis (e.g., paired $t$-test or Wilcoxon signed-rank test) requires more independent repetitions, we plan to include such detailed significance testing—reporting $p$-values and effect sizes—in future work. This will provide further quantitative confirmation of the observed trends.

## 6. Conclusion

This study proposed the Multi-Order Hyperbolic Graph Convolution with Aggregated Attention (MOHGCAA) framework to address the complex challenges of social event detection (SED). By integrating multi-order graph modeling with

curvature-aware attention mechanisms, our method effectively captures both hierarchical and high-order dependencies inherent in social event data. The design allows for a more faithful representation of real-world event relationships that are often hierarchical, imbalanced, and dynamically evolving.

Extensive experiments across multiple benchmark datasets demonstrated the superior performance of MOHGCAA under both supervised and unsupervised settings. The results confirmed that the proposed hyperbolic aggregation mechanism achieves consistent gains in accuracy, stability, and generalization, outperforming existing Euclidean and hyperbolic counterparts. These findings underscore the robustness and adaptability of the proposed approach, highlighting its capability to enhance downstream SED applications in domains such as crisis monitoring, news trend analysis, and social media intelligence.

Despite these promising results, several limitations remain. The current design relies on tangent-space projections to perform convolution and optimization, which may introduce local geometric distortion when modeling deep hierarchies or highly curved manifolds. Future extensions could explore direct learning within non-tangent hyperbolic manifolds, enabling more expressive representations without repeated mapping operations. In addition, computational costs increase with graph size, potentially limiting scalability to very large dynamic networks. Subsequent research may focus on efficient curvature-adaptive sampling, model interpretability through geometric visualization, and integration with temporal reasoning for real-time event detection. These directions will further advance hyperbolic graph frameworks toward more general non-Euclidean learning paradigms.

## Author contributions

**Conceptualization:** Tien Ping Tan.

**Data curation:** Zhilan Liu, Yuxin Li.

**Investigation:** Zhilan Liu, Yuxin Li.

**Methodology:** Yao Liu.

**Supervision:** Tien Ping Tan.

**Visualization:** Zhilan Liu.

**Writing – original draft:** Yao Liu.

**Writing – review & editing:** Yao Liu.

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
