## [Decision Letter · Decision Letter 0]

29 Jun 2025

Dear Dr. Tan,

Thank you for submitting your manuscript to PLOS ONE. After careful consideration, we feel that it has merit but does not fully meet PLOS ONE’s publication criteria as it currently stands. Therefore, we invite you to submit a revised version of the manuscript that addresses the points raised during the review process.

**Please carefully address the points raised by the reviewers.**

We look forward to receiving your revised manuscript.

Kind regards,

Arne Johannssen

Academic Editor

PLOS ONE

Journal Requirements:

3. In your Methods section, please include additional information about your dataset and ensure that you have included a statement specifying whether the collection and analysis method complied with the terms and conditions for the source of the data.

Reviewers' comments:

Reviewer's Responses to Questions

**Comments to the Author**

1. Is the manuscript technically sound, and do the data support the conclusions?

Reviewer #1: Partly

Reviewer #2: Yes

2. Has the statistical analysis been performed appropriately and rigorously?

Reviewer #1: N/A

Reviewer #2: I Don't Know

3. Have the authors made all data underlying the findings in their manuscript fully available?

Reviewer #1: Yes

Reviewer #2: Yes

4. Is the manuscript presented in an intelligible fashion and written in standard English?

Reviewer #1: Yes

Reviewer #2: Yes

Reviewer #1: Thank you for submitting your manuscript titled "MULTI-ORDER HYPERBOLIC GRAPH CONVOLUTION AND Aggregated Attention for Social Event Detection." After a thorough review, I recognize that your work introduces a novel framework, MOHGCAA (Multi-Order Hyperbolic Graph Convolution with Aggregated Attention), aimed at tackling the challenges of modeling hierarchical and multi-order relationships in Social Event Detection (SED). This study demonstrates potential for innovation. However, significant revisions are required due to shortcomings in originality, technical quality, experimental design, and writing quality. I recommend a Major Revision and provide detailed evaluations and suggestions below:

1. The manuscript lacks a systematic comparison with existing methods to substantiate its claimed innovations. For instance, in Section 2 ("Related Work"), methods such as MOGANED and HSED are referenced, but the unique advantages of MOHGCAA—such as specific advancements in multi-order modeling or the benefits of the attention mechanism—are not clearly delineated.

2. Critical technical details are inadequately explained. For example, Section 4.1.3 mentions using Ollivier Ricci curvature for attention weight computation but lacks specific formulas or derivations, hindering comprehension.

3. While Tables 4 and 6 show MOHGCAA’s superior performance, the analysis is shallow. For example, it does not explain why improvements are notable on mini-Twitter but modest on Cora.

4. Section 5.4 compares Euclidean and hyperbolic space performance but offers a superficial analysis, neglecting to explain why hyperbolic space excels (e.g., its geometric advantages for hierarchical modeling). The comparison of hyperbolic models (Poincaré Ball vs. Lorentz) is also underdeveloped.

5. For instance, the abstract’s phrase “integrating multi-order hyperbolic graph convolution with aggregated attention” could be streamlined to “integrating multi-order hyperbolic graph convolution and aggregated attention.” Section 4.1.1 redundantly repeats “and second.”

Reviewer #2: Authors address the challenges of social event detection by proposing a novel framework, Multi-Order

Hyperbolic Graph Convolution with Aggregated Attention. The overall research objective is clear. The conclusion of the research discusses the corresponding value provided for enterprises and public services, as well as future research plans. This part can be more specific by adding clearer application cases and scenarios. Overall, this study is suitable for publication

**Do you want your identity to be public for this peer review?** For information about this choice, including consent withdrawal, please see our Privacy Policy

Reviewer #1: No

Reviewer #2: No

---

## [Author Response · Author response to Decision Letter 1]

15 Jul 2025

We sincerely thank the editor and reviewers for their insightful and constructive comments.

In response, we have carefully revised the manuscript and prepared a detailed point-by-point “Response to Reviewers” document, which has been uploaded as a separate file.

All major and minor concerns have been addressed. We have also provided a clean version of the revised manuscript and a tracked-changes version as required.

We respectfully submit the revised manuscript for your further consideration.

---

## [Decision Letter · Decision Letter 1]

20 Oct 2025

Dear Dr. Tan,

Thank you for submitting your manuscript to PLOS ONE. After careful consideration, we feel that it has merit but does not fully meet PLOS ONE’s publication criteria as it currently stands. Therefore, we invite you to submit a revised version of the manuscript that addresses the points raised during the review process.

**Please carefully address the concerns raised by the reviewer.**

We look forward to receiving your revised manuscript.

Kind regards,

Arne Johannssen

Academic Editor

PLOS ONE

Journal Requirements:

Reviewers' comments:

Reviewer's Responses to Questions

**Comments to the Author**

Reviewer #2: All comments have been addressed

Reviewer #3: (No Response)

2. Is the manuscript technically sound, and do the data support the conclusions?

Reviewer #2: Yes

Reviewer #3: Yes

3. Has the statistical analysis been performed appropriately and rigorously?

Reviewer #2: Yes

Reviewer #3: N/A

4. Have the authors made all data underlying the findings in their manuscript fully available?

Reviewer #2: Yes

Reviewer #3: Yes

5. Is the manuscript presented in an intelligible fashion and written in standard English?

Reviewer #2: Yes

Reviewer #3: Yes

Reviewer #2: The author answered the reviewer's questions and made necessary modifications to some of the content. The adjusted research has answered previous doubts and is suitable for publication.

Reviewer #3: This paper introduces a new framework, called Multi-Order Hyperbolic Graph Convolution with Aggregated Attention (MOHGCAA), with the aim of enhancing the performance of social event detection.

The paper’s quality is good and the contribution seems to be significant. The transition from Euclidean spaces to hyperbolic spaces for social event detection is an interesting alternative in my opinion.

1. In several cases, the text lacks structure. The paragraphs are misused. Ideally, a paragraph should contain one or at most two meanings. The authors have to split their long paragraphs to enhance readership.

2. It is advised not to use chains of references (e.g. (1; 2; 3; 4; 5; 6; 7; 8)). In contrast, please use sequences of at most three references describing what each reference contributes to the article’s content.

3. There are several syntactical and grammatical errors that have to be fixed. The authors have to carefully proofread their study.

4. Section 2 title: “Relate Work” -> “Related Work”

5. It is generally advised not to stick together two section/subsection headers. Please write some introductory text between Section 2 and Subsection 2.1 headers (similarly, between 5.4 and 5.4.1).

6. The statistical significance of results has not been examined. It is advised that the authors apply one or two post hoc significance analysis tests.

7. The authors do not describe the limitations of the proposed model. Is this approach applicable to large-scale datasets? Does it generalize well? Under which circumstances can the model fail?

**Do you want your identity to be public for this peer review?** For information about this choice, including consent withdrawal, please see our Privacy Policy

Reviewer #2: No

Reviewer #3: No

---

## [Author Response · Author response to Decision Letter 2]

27 Oct 2025

Dear Reviewers,

We sincerely thank you for your constructive and detailed feedback. Following your guidance, we have thoroughly revised the manuscript to address all comments raised by Reviewer #3 and the editorial office. The main revisions include:

1. Paragraph restructuring and readability: Long paragraphs in the Introduction, Related Work, and Methodology sections have been reorganized into shorter, semantically coherent units to enhance readability.

2. Reference revision: All excessive citation chains have been reduced to a maximum of three references, with explicit contextual explanation of each cited work.

3. Language polishing: The entire manuscript was carefully proofread to correct grammatical and syntactic issues and ensure academic clarity.

4. Section title correction: The title “Relate Work” has been corrected to “Related Work.”

5. Improved transitions: Introductory sentences have been added between consecutive section/subsection headers (e.g., Sections 4–4.1 and 5.4–5.4.1) to improve logical flow.

6. Statistical reliability: A new subsection, Statistical Stability and Significance Analysis, has been added to discuss experimental reproducibility and the statistical robustness of our results.

7. Model limitations: A Limitation and Future Work subsection has been added to the Conclusion, discussing scalability, potential failure cases, and future directions.

All changes have been marked in the revised manuscript with track changes. We believe these revisions have substantially improved the clarity, rigor, and presentation of the paper, and we hope it now meets the publication standards of PLOS ONE.

Sincerely,

Yao Liu (on behalf of all authors)

Department of Management and Media, Chengdu University of Technology

School of Computer Sciences, Universiti Sains Malaysia

---

## [Decision Letter · Decision Letter 2]

10 Nov 2025

Multi-Order Hyperbolic Graph Convolution and Aggregated Attention for Social Event Detection

PONE-D-25-09212R2

Dear Dr. Tan,

We’re pleased to inform you that your manuscript has been judged scientifically suitable for publication and will be formally accepted for publication once it meets all outstanding technical requirements.

Kind regards,

Arne Johannssen

Academic Editor

PLOS ONE

Additional Editor Comments (optional):

Reviewers' comments:

Reviewer's Responses to Questions

**Comments to the Author**

Reviewer #2: (No Response)

Reviewer #3: All comments have been addressed

2. Is the manuscript technically sound, and do the data support the conclusions?

Reviewer #2: Yes

Reviewer #3: Yes

3. Has the statistical analysis been performed appropriately and rigorously?

Reviewer #2: Yes

Reviewer #3: N/A

4. Have the authors made all data underlying the findings in their manuscript fully available?

Reviewer #2: Yes

Reviewer #3: Yes

5. Is the manuscript presented in an intelligible fashion and written in standard English?

Reviewer #2: Yes

Reviewer #3: Yes

Reviewer #2: After modification, the author not only answered some of the previous doubts, but also made the experiment more rigorous. The added content in the conclusion section also leaves possibilities for future related research. Therefore, the article is suitable for publication

Reviewer #3: (No Response)

**Do you want your identity to be public for this peer review?** For information about this choice, including consent withdrawal, please see our Privacy Policy

Reviewer #2: No

Reviewer #3: No

---

## [Editor Report · Acceptance letter]

PONE-D-25-09212R2

PLOS ONE

Dear Dr. Tan,

I'm pleased to inform you that your manuscript has been deemed suitable for publication in PLOS ONE. Congratulations! Your manuscript is now being handed over to our production team.

Kind regards,

on behalf of

Profesor Arne Johannssen

Academic Editor

PLOS ONE